# Utilizing Speaker Profiles for Impersonation Audio Detection

## ABSTRACT

Fake audio detection is an emerging active topic. A growing number of literatures have aimed to detect fake utterance, which are mostly generated by Text-to-speech (TTS) or voice conversion (VC). However, countermeasures against impersonation remain an underexplored area. Impersonation is a fake type that involves an imitator replicating specific traits and speech style of a target speaker. Unlike TTS and VC, which often leave digital traces or signal artifacts, impersonation involves live human beings producing entirely natural speech, rendering the detection of impersonation audio a challenging task. Thus, we propose a novel method that integrates speaker profiles into the process of impersonation audio detection. Speaker profiles are inherent characteristics that are challenging for impersonators to mimic accurately, such as speaker's age, job. We aim to leverage these features to extract discriminative information for detecting impersonation audio. Moreover, there is no large impersonated speech corpora available for quantitative study of impersonation impacts. To address this gap, we further design the first large-scale, diverse-speaker Chinese impersonation dataset, named ImPersonation Audio Detection (IPAD), to advance the community's research on impersonation audio detection. We evaluate several existing fake audio detection methods on our proposed dataset IPAD, demonstrating its necessity and the challenges. Additionally, our findings reveal that incorporating speaker profiles can significantly enhance the model's performance in detecting impersonation audio.

## CCS CONCEPTS

• **Security and privacy** → **Spoofing attacks**; **Biometrics**.

## KEYWORDS

Fake Audio Detection, Impersonation Audio Dataset, Speaker Profiles

## 1 INTRODUCTION

Over the past few years, speech synthesis and voice conversion technologies have made great improvement, enabling the generation of high-fidelity and human-like speech [29, 44]. However, the misuse of these technologies can facilitate the spread of misleading information and contribute to cybercrimes such as fraud and extortion [42]. Given the devastating consequences of fake audio, fake audio detection has become an urgent and essential task that needs to be addressed.

*ACM MM, 2024, Melbourne, Australia*
© 2024 Copyright held by the owner/author(s). Publication rights licensed to ACM.
ACM ISBN 978-x-xxxx-xxxx-x/YY/MM
https://doi.org/10.1145/nnnnnnn.nnnnnnn

In recent years, a growing number of scholars have made attempts to detect fake audio. Most previous fake audio detection research has primarily focused on four kinds of fake types: text-to-speech, voice conversion, replay and partially fake [42]. Text-to-speech (TTS) [34] is a technique that generates intelligible and natural speech from any given text via deep learning based models. Voice conversion (VC) [34] aims to alter the timbre and prosody of a given speaker's speech to that another speaker, while keeping the content of the speech remains the same. These two spoofing techniques are widely used in a series of competitions, such as the ASVspoof [31] and ADD challenge [40, 41]. Replay attack [17] is referred to as a form of replaying pre-recorded genuine utterances of a target speaker to an automatic speaker verification (ASV) system. Partially fake [39] focuses on only changing several words in an utterance. where fake segment is generated by manipulating the original utterances with genuine or synthesized audio clips. Despite the considerable attention given to these spoofing techniques [9], countermeasures against impersonation remain relatively underexplored [25, 34].

Impersonation [9, 25, 34] entails an imitator mimicking specific traits associated with the prosody, pitch, dialect, lexical and speech style of a particular target speaker. This form of fake audio is generated by real human beings and poses a significant threat to speaker verification systems, as criminals could potentially use impersonation to gain unauthorized access [25]. Also, Wu et al. [34] points out that, despite its higher cost, impersonation audio is more effective at evading detection due to its naturalness, making it a challenging fake type to detection, when compared to TTS and VC.

Unlike the four previously mentioned spoofing attacks, which typically leave traces via the physical characteristics of recording and playback devices, or through artifacts introduced by signal processing in synthesis or conversion systems, impersonation audio is entirely natural speech produced by actual human beings [25, 34]. This makes the detection of impersonation audio a challenging task. Thus we propose an innovative method that integrates the speaker profiles into the detection of impersonation audio. Speaker profiles refer to inherent attributes such as the speaker's age, hometown, job and so on. We aim to leverage these inherent characteristics that are challenging for impersonators to imitate accurately for impersonation audio detection. As speakers from the same hometown typically share accents and those with the same job often use a similar lexicon, a graph-based approach is ideal for modeling the interconnected relationships between these attributes [22, 27]. Accordingly, we introduce a speaker profile extractor that employs a mutual information-based graph embedding method [22, 27] to gather speaker profile information. Subsequently, the features enriched with the speaker profiles are integrated with the features derived from the front-end feature extractor module through a fusion module. Finally, the fusion module's output is fed to the back-end classifier, which generates the high-level representation aiming at distinguishing between impersonated utterances and genuine ones.

| Name | Year | Language | Fake Types | Traits | # Utts | # Hours | # Spks |
|---|---|---|---|---|---|---|---|
| FoR [23] | 2019 | English | TTS | Clean | 195,541 | 150.3 | Fake:33/Real:140 |
| ASVspoof 2021 [31] | 2021 | English | TTS, VC, Replay | Noisy | 1,566,273 | 325.8 | Fake:133/Real:133 |
| In-the-Wild [20] | 2022 | English | TTS | Social Media | 31,779 | 38.0 | Fake:58/Real:58 |
| ADD 2022 [40] | 2022 | Chinese | TTS, VC, Partially Fake | Noisy | 493,123 | - | - |
| ADD 2023 [41] | 2023 | Chinese | TTS, VC, Partially Fake | Noisy | 517,068 | - | - |
| IPAD | 2024 | Chinese | Impersonation | Web Media | 24,074 | 23.5 | Fake:408/Real:258 |

**Table 1: Characteristics of representative datasets on fake audio detection. # Utts, # Hours and # Spks represent number of utterances, hours and speakers, respectively. We will make our dataset publicly available once our paper is accepted.**

However, a significant obstacle is the absence of large-scale impersonated speech datasets, limiting quantitative analysis of impersonation effects, primarily due to the challenges associated with acquiring high-quality impersonation data, which is both scarce and expensive. An impersonation dataset is designed by [18], focusing on investigating the vulnerability of speaker verification. However, only two inexperienced impersonators are involved to mimic utterances from YOHO corpus [4]. Hautamäki et al. [9] constructs a small Finnish impersonation dataset in 2013. All these prior datasets suffer limitations such as few speakers and short durations. Addressing these gaps, this paper presents a diverse-scenarios, diverse-speaker impersonation dataset, named ImPersonation Audio Detection (IPAD), to benefit the community's research. As Sahidullah et al. [25] indicate that professional impersonators result in higher deception rates than amateurs, we specifically curated our dataset with audio from skilled impersonators, making our IPAD dataset more pratical. Moreover, we are committed to ensuring a balanced distribution of speakers across different age groups and genders.

The main contributions of this paper are as follows:

- We propose a novel method that integrates speaker profiles into the detection of impersonation audio. To this end, we utilize a graph-based approach to extract speaker profile information. Additionally, our proposed method does not require labeled speaker profiles during the test period.
- We present the first diverse-scenarios, diverse-speaker impersonation dataset, named ImPersonation Audio Detection (IPAD), to promote the community's research on impersonation audio detection. The impersonation dataset will be publicly available.
- We perform comprehensive baseline benchmark evaluation and demonstrated our speaker profiles integrated method can achieve impressive results.

## 2 RELATED WORK

### 2.1 Fake Audio Detection Methods

In recently years, many detection methods have been introduced to discriminate fake audio files from real speech, mainly focusing on the pipeline detector and end-to-end detector solutions [42].

The feature extraction, which aims to learn discriminative features via capturing audio fake artifacts from speech signals, is the key module of the pipeline detector. The features used in previous can be roughly divided into two categories [42]: handcrafted

features and deep features. Linear frequency cepstral coefficients (LFCC) is a commonly used handcrafted features that uses linear filerbanks, capturing more spetral details in the high frequency region. LFCC in conjunction with Gaussian Mixture Models (GMM) and Light Convolutional Neural Networks (LCNN), have been adopted as the baseline models for ASVspoof 2021 [31] and ADD challenge. [40, 41]. Nevertheless, handcrafted features are flawed by biases due to limitation of handmade representations [43]. Deep features, derived from deep neural networks, have been proposed to address these limitations. Pre-trained self-supervised speech models, such as Wav2vec [3], Hubert [11] and WavLM [5], are the most widely used ones [30]. Wang and Yamagishi [30] investigate the performance of spoof speech detection using embedding features extracted from different self-pretrained models. The back-end classifier, tasked with learning high-level feature representations from the front-end input features, is indispensable in the fake audio detection. One of the extensively used classifiers is Light CNN (LCNN) [33], as it is an effective model employed as the baseline model in a series of competitions, such as ASVspoof 2017 [17], ASVspoof 2019 [21] and ADD 2022 [40].

End-to-End Models are deep neural networks that integrate feature extraction and classification in an end-to-end manner have shown competitive performance in fake audio detection. Notable models include RawNet2 [14] and its derivatives, RawNet3 [15] and TO-RawNet [28]; the Differentiable Architecture Search (DARTS) influenced Raw PC-DARTS [8]; Transformer-based Rawformer [37]; the Graph Neural Network-based AASIST [13] and its orthogonal regularization variant, Orth-AASIST [28];.

However, previous models have primarily targeted fake types such as TTS and VC, which often leave digital traces or signal artifacts. In contrast, impersonation audio is entirely natural speech produced by real humans, making it challenging for these methods to effectively detect. Filling this gap, in this paper, we propose a novel method that integrates speaker profiles that are inherent characteristics that are challenging for impersonators to mimic accurately into the detection of impersonation audio.

### 2.2 Fake Audio Detection Datasets

The advancement of fake audio detection techniques significantly depends on well-established datasets, which encompass various fake types and diverse acoustic conditions. Table 1 summarizes the characteristics of representation datasets in the field of fake audio detection along with our proposed dataset.

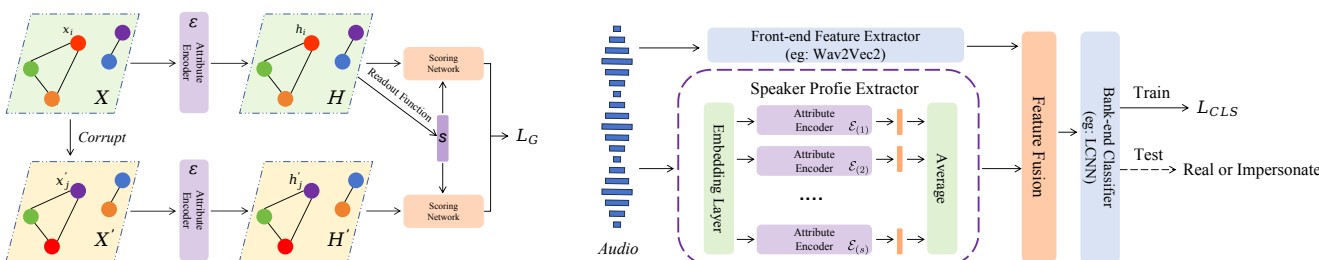

**Figure 1: Overview of the training process of the attribute encoder (left figure) and detail for speaker profiles integrated framework (right figure).**

| Dataset | # Target | # Imitators | Professional |
|---|---|---|---|
| Lau et al. [18] | 6 | 2 | No |
| Farrús Cabeceran et al. [6] | 5 | 2 | Yes |
| Hautamäki et al. [9] | 5 | 1 | Yes |
| IPAD | 799 | 408 | Yes |

**Table 2: Summary of impersonation spoofing attack dataset. # Target and # Imitators represent number of target speakers and impersonators. Professional represents whether the audio is imitated by professional impersonators.**

Most earlier spoofed datasets were primarily developed to bolster defenses against spoofing attacks in ASV systems. Moreover, the spoofing types are not diverse. Some spoofing datasets focus exclusively on a single type of TTS method [19] or a specific VC Method [1]. To alleviate this issue, Wu et al. [35] design a standard public spoofing dataset SAS which consists of various TTS and VC methods. The SAS dataset is used to support ASVspoof 2015 [36], which aims to detect the spoofed speech. Replay is considered as a low cost and challenging attack included in the ASVspoof 2017 challenge [17]. The ASVspoof 2019 [21] and 2021 datasets [31] both consist of replay, TTS and VC attacks.

In recent years, a few attempts have been made to design datasets mainly for fake audio detection systems. In 2020, Reimao and Tzerpos [23] developed a publicly available dataset FoR containing synthetic utterances, which are generated with open-source TTS tools. In 2021, Frank and Schönherr [7] developed a fake audio dataset named WaveFake, which contains two speaker's fake utterances synthesised by the latest TTS models. However, these datasets have not covered some real-life challenging situations. The datasets in ADD 2022 challenge [40] are designed to fill the gap. The fake utterances in LF dataset are generated using the latest state-of-the-art TTS and VC models, which contain diversified noise interference. The fake utterances in PF dataset are chosen from the HAD dataset [39] designed by, which are generated by manipulating the original genuine utterances with real or synthesized audio segments.

Few previous studies have been dedicated to the construction of voice imitation datasets. In 2004, an impersonation database is developed by [18], which is used for investigating the vulnerability of speaker verification. Two novice impersonators were tasked with mimicking voices from the YOHO corpus. They listened to and

subsequently imitated 40 training utterances from selected speakers. In 2013, a small Finnish impersonation dataset was designed by [9]. Our dataset IPAD markedly distinguishes itself from previous efforts by incorporating a significantly larger number of speakers. Additionally, our audio is extracted from videos downloaded from entertainment programs on web media, enhancing the practicality of our IPAD dataset.

## 3 METHOD

In this section we provide details of the developed methods to detect impersonation audio. An overview of our approach is illustrated in Figure 1.

### 3.1 Attribute Encoder

Speaker profiles include different attributes, such as speaker's age, hometown. For each attribute, we will learn a attribute encoder to extract attribute-specific information.

Speakers with the same attribute value often exhibit similar characteristics. For instance, speakers from the same hometown typically share similar accents. Thus, we employ a graph-based approach, in which we model each audio in the training set as a node, to effectively model the interconnected relationships between speaker attribute.

We first give a introduce to the problem statement. Suppose we are provided with a set of node features, in our method, i.e. audio features, $\mathbf{X} = \{\mathbf{x}_1, \mathbf{x}_2, \ldots, \mathbf{x}_N\}$, where $N$ is the number of nodes in the graph and $\mathbf{x}_i \in \mathbb{R}^f$ encodes the feature of node $i$. We are also provided with relational information between these nodes in the form of an adjacency matrix, $\mathbf{A} \in \mathbb{R}^{N \times N}$. We assume that $A_{ij} = 1$ if there exists an edge between node $i$ and node $j$, for example, node $i$ and node $j$ have the same job in job attribute encoder. We draw inspiration from [22, 27], where learn the encoder relaying on maximizing local mutual information between global summary vector and local node representations. More precisely, we learn a low-dimensional representation for each node $\mathbf{x}_i$, i.e., $\mathbf{h}_i \in \mathbb{R}^d$, such that the average mutual information between the global summary vector $\mathbf{s} \in \mathbb{R}^d$, and local node representations $\{\mathbf{h}_1, \mathbf{h}_2, \ldots, \mathbf{h}_N\}$ is maximized.

To this end, we first introduce a attribute encoder $\mathcal{E}$, consisting two linear layer with ReLU activation. Then we can generate the local node representation matrix $\mathbf{H}$ following Eq. (1)

$$H = \sigma \left( \hat{D}^{-\frac{1}{2}} \hat{A} \hat{D}^{-\frac{1}{2}} \mathcal{E}(X) \right) \quad (1)$$

where $\hat{A} = A + wI_N$, $I_N$ is the $N \times N$ identity matrix. $\hat{D}_{ii} = \sum_j \hat{A}_{ij}$, $\mathcal{E}$ is the trainable network, and $\sigma$ is the ReLU non-linearity function. Our approach adjusts the impact of self-connections by introducing a weight parameter $w \in \mathbb{R}$. A higher $w$ value increases the node's self-relevance in its embedding, consequently lessening the influence of adjacent nodes.

Then we can calculate the graph-level summary representation $s$ by employing a readout function $\mathcal{R}$.

$$s = \mathcal{R}(H) = \sigma \left( \frac{1}{N} \sum_{i=1}^{N} h_i \right) \quad (2)$$

where $\sigma$ is the logistic sigmoid non-linearity function, and $h_i$ represents the $i$-th row of the node embedding matrix $H$.

We follow [22, 27], introducing a scoring network $\mathcal{D}$ that discriminates the true samples. i.e., $(h_i, s)$ from $(h'_j, s)$ as a proxy for maximizing the local mutual information. $\mathcal{D}((h_i, s))$ represents the probability scores assigned to the patch-summary pair. Here negative representation $h'_j$ is obtained by row-wise shuffling, i.e. $X \rightarrow X'$. The corruption function used here is designed to encourage the representations to properly encode structural similarities of different nodes. Then we calculate the $H'$ following Eq.(1). The scoring network scores patch-summary pairs by applying a simple bilinear transformation function:

$$\mathcal{D}(h_i, s) = \sigma \left( h_i^T M s \right) \quad (3)$$

where $\sigma$ is the logistic sigmoid non-linearity function, and $M$ is the trainable scoring matrix.

Finally, we can update parameters of $\mathcal{E}$, $\mathcal{R}$ and $\mathcal{D}$ by optimizing the following attribute specific cross entropy loss $\mathcal{L}_G$.

$$\mathcal{L}_G = \sum_{i=1}^{N} \log \mathcal{D}(h_i, s) + \sum_{j=1}^{N} \log(1 - \mathcal{D}(h_j, s)) \quad (4)$$

## 3.2 Framework of our proposed method

In this subsection, we will describe the framework of our proposed speaker profiles integrated detection method in detail.

For each attribute, we first train an attribute encoder to extract attribute specific information following the method described in subsection 3.1. The input audio features are obtained by passing the audio through a Embedding layer. Suppose we have $s$ types of attributes, we can obtain $s$ attribute encoders $\{\mathcal{E}_{(1)}, \mathcal{E}_{(2)}, \ldots, \mathcal{E}_{(s)}\}$. There encoders contain relevant information regarding corresponding speaker profile.

Suppose we are provided a mini-batch of audio, $W = \{w_1, w_2, \ldots, w_n\}$. We first employ the speaker profile extractor to extract speaker profile information. Specifically, for each $w_i$, we adopt an Embedding layer $E$, followed by $s$ learned attribute encoders to obtain the speaker profiles incorporated representations, then we take an average of these representations:

$$v_i = Avg \left\{ \mathcal{E}_{(1)}[E(w_i)], \mathcal{E}_{(2)}[E(w_i)], \ldots, \mathcal{E}_{(s)}[E(w_i)] \right\} \quad (5)$$

Here $Avg$ represents the averaging operation. The $v_i$ calculated by Eq.(5) is the output of the speaker profile extractor for $w_i$. A key consequence is that the produced representation $v_i$ contains speaker profile information, such as speaker's age, hometown, job. Then we can obtain $V = \{v_1, v_2, \ldots, v_n\}$.

Next, we can derive the deep features $Q$ by employing the front-end feature extractor $\mathcal{F}_{extractor}$ (wav2vec2 [3] for example).

$$Q = \{q_1, q_2, \ldots, q_n\} = \mathcal{F}_{extractor}(W) \quad (6)$$

Subsequently, We amalgamate representation $V$ from the speaker profile extractor and $Q$ from the front-end feature extractor via a feature fusion module $\mathcal{F}_{fusion}$.

$$K = \{k_1, k_2, \ldots, k_n\} = \mathcal{F}_{fusion}(V, Q) \quad (7)$$

Here $K$ represents the integrated representation. This is obtained by the frame-level concatenation of corresponding features from $V$ and $Q$. In detail, if $q_i$ is the feature with dimensions $(t, f)$, where $t$ represents the number of time frames and $f$ represents the embedding size of the frond-end feature extractor. $v_i$ is the feature with dimensions $(l, )$, where $l$ denotes the output size of the speaker profile extractor, then $v_1$ is replicated $m$ times to align with the dimensions of $q_1$. As a result, the fused feature $k_1$ will have dimensions $(t, f + l)$.

Ultimately, we engage a back-end classifier $\mathcal{F}_{classifier}$, Light CNN (LCNN) [33] for example, to detect fake audio. Suppose the labels for the mini-batch of audio $W$ are $Y = \{y_1, y_2, \ldots, y_n\}$. We can formulate the classification loss $\mathcal{L}_{CLS}$ as follows:

$$\mathcal{L}_{CLS} = -\frac{1}{n} \sum_{i=1}^{n} \left[ y_i \log(\hat{y}_i) + (1 - y_i) \log(1 - \log(\hat{y}_i)) \right] \quad (8)$$

where $\hat{y}_i$ is the model's predicted probability that the $i$-th audio is a bona fide one. Here the prediction is obtained by feeding $K$ to $\mathcal{F}_{classifier}$. Then we can update the parameters of $\mathcal{F}_{classifier}$ by optimizing $\mathcal{L}_{CLS}$.

Additionally, during the test phase, since we have already trained the attribute encoder for each speaker profile, our method does not require labeled speaker profiles for prediction.

# 4 DATASET

## 4.1 Dataset Collection Policy

We construct our impersonation dataset IPAD through five steps. The audio is extracted from videos downloaded from web media, resulting in the "in the wild" characteristics of our IPAD dataset.

*Step 1: Download Videos.* In order to build our dataset from scratch, we collect videos from several variety entertainment programs that contain segments featuring impersonators mimicking others. The programs include The Sound[1], Voice Monster[2], Voice Acting's Influence[3], Lucky Start[4], Cheerful Gathering[5], Tu cara me

---

[1] https://zh.wikipedia.org/wiki/%E5%A3%B0%E4%B8%B4%E5%85%B6%E5%A2%83
[2] https://zh.wikipedia.org/wiki/%E6%88%91%E6%98%AF%E7%89%B9%E4%BC%98%E5%A3%B0
[3] https://baike.baidu.com/item/%E5%A3%B0%E6%BC%94%E7%9A%84%E5%8A%9B%E9%87%8F/57929334
[4] https://zh.wikipedia.org/wiki/%E5%BC%80%E9%97%A8%E5%A4%A7%E5%90%89
[5] https://baike.baidu.com/item/%E6%AC%A2%E4%B9%90%E6%80%BB%E5%8A%A8%E5%91%98/11011573

| | Real | | | Fake | | | Total | | |
|---|---|---|---|---|---|---|---|---|---|
| | # Utts | # Spks | # Hours | # Utts | # Spks | # Hours | # Utts | # Spks | # Hours |
| Train | 1,400 | 58 | 1.9 | 2,893 | 68 | 2.9 | 4,293 | 73 | 4.8 |
| Dev | 747 | 37 | 0.9 | 1,775 | 38 | 1.8 | 2,522 | 43 | 2.7 |
| Test | 1,909 | 78 | 2.6 | 4,558 | 78 | 4.6 | 6,497 | 95 | 7.2 |
| Unseen | 1,114 | 138 | 1.6 | 9,648 | 277 | 7.2 | 10,762 | 296 | 8.8 |

Table 3: Key statistics for the IPAD dataset. It consists of four sets: train, dev, test and unseen test (unseen) sets. We enumerates the number of utterances (# Utts), speakers (# Spks), and total hours (# Hours) for each subset, with an additional column summarizing the combined totals.

| Scenarios | | | Dubbing | | | | | Conversational Speech | | | | |
|---|---|---|---|---|---|---|---|---|---|---|---|---|
| | | | # Utts | # Spks | # HTs | # Ages | # Jobs | # Utts | # Spks | # HTs | # Ages | # Jobs |
| Train | Real | Male | 1,157 | 36 | 14 | 21 | 4 | 19 | 7 | 7 | 6 | 5 |
| | | Female | 221 | 13 | 7 | 12 | 2 | 3 | 3 | 3 | 3 | 3 |
| | Fake | Male | 1,893 | 40 | 15 | 27 | 5 | 261 | 8 | 6 | 7 | 6 |
| | | Female | 689 | 15 | 9 | 16 | 2 | 50 | 6 | 4 | 6 | 5 |
| Dev | Real | Male | 593 | 21 | 10 | 15 | 4 | 120 | 8 | 6 | 7 | 3 |
| | | Female | 120 | 8 | 6 | 7 | 3 | 7 | 1 | 1 | 1 | 1 |
| | Fake | Male | 1,180 | 22 | 10 | 18 | 4 | 106 | 5 | 5 | 5 | 4 |
| | | Female | 475 | 10 | 7 | 9 | 3 | 14 | 1 | 1 | 1 | 1 |
| Test | Real | Male | 1,475 | 47 | 18 | 27 | 4 | 80 | 17 | 11 | 12 | 11 |
| | | Female | 331 | 9 | 5 | 9 | 2 | 23 | 6 | 5 | 4 | 6 |
| | Fake | Male | 3,009 | 53 | 19 | 31 | 5 | 360 | 10 | 8 | 7 | 4 |
| | | Female | 1,118 | 14 | 8 | 14 | 4 | 101 | 3 | 2 | 3 | 3 |
| Scenarios | | | Singing | | | | | Other | | | | |
| Unseen | Real | Male | 592 | 82 | 26 | 32 | 21 | 195 | 15 | 6 | 5 | 4 |
| | | Female | 302 | 38 | 18 | 20 | 10 | 25 | 6 | 5 | 5 | 1 |
| | Fake | Male | 5,804 | 185 | 34 | 39 | 30 | 107 | 4 | 3 | 3 | 3 |
| | | Female | 3,723 | 92 | 27 | 30 | 16 | 14 | 1 | 1 | 1 | 1 |

Table 4: Detailed statistics for the real utterances and fake utterances in our IPAD dataset. # Utts, # Spks, # HTs, # Ags, # Jobs represent number of utterances, speakers, hometowns, ages and jobs, respectively.

suena [6], Fun with Liza and Gods [7], Copycat Singers[8]. In total, we have collected 168.34 hours of video for subsequent audio slicing in the imitation dataset.

*Step 2: Manual Labeling.* We recruit nine annotators to label our dataset. For each video, they are tasked with identifying segments where the impersonator is speaking as themselves and segments where the impersonator is mimicking others, marking the start and end times with precision to the second for subsequent audio segmentation. For the first condition, annotations required include the speaker's name, hometown, age, job, gender, and the scenario. In instances of imitation, annotations needed to cover the impersonator's name, hometown, age, job, gender, the scenario of the imitation, as well as the name of the person being imitated. For the hometown, we require annotations to be specific to the province level. Regarding the scenario, they were categorized into **dubbing**,

conversational speech, and **singing**. If a segment did not fit into these three categories, it was labeled as '**other**'. For virtually all videos, two annotators were assigned to provide labels. Subsequently, a reviewer would reconcile any discrepancies between the annotations, making necessary adjustments. This process was instituted to ensure the quality of the dataset.

*Step 3: Extract Audio from Video.* we leverage FFmpeg [9] to extract and convert specific audio segments from videos into mono wav files with a 16,000 Hz sampling rate. This process ensures the standardization of our audio dataset for consistent analysis.

*Step 4: Audio Segmentation.* Following the extraction of the audio, we employed a Voice Activity Detection (VAD) tool [38] to eliminate segments of silence. For audio clips exceeding 10 seconds in duration, the VAD model[10] [38] was utilized to determine the start and end points of valid speech within the input audio, ultimately discarding non-speech parts in the audio.

---

[6] https://zh.wikipedia.org/wiki/%E7%99%BE%E5%8F%98%E5%A4%A7%E5%92%96%E7%A7%80

[7] https://zh.wikipedia.org/wiki/%E8%8D%83%E5%8A%A0%E7%A6%8F%E7\%A5%BF%E5%A3%AB

[8] https://baike.baidu.com/item/%E5%A4%A9%E4%B8%8B%E6%97%A0%E5%8F%8C/19885272

[9] https://ffmpeg.org/

[10] https://modelscope.cn/models/iic/speech_fsmn_vad_zh-cn-16k-common-pytorch/summary

*Step 5: Train, Dev, Test and Unseen Test Split.* After acquiring the complete set of audio, we split the audio from dubbing and conversational speech scenarios into train, dev, and test sets. The allocation is based on the number of utterances per speaker, with the stipulation that the speakers across the train, dev, and test sets must be mutually exclusive to avoid any overlap. In detail, to ensure the dataset's train, dev, and test splits maintain balance in terms of age and gender, we divide speakers into four age groups: 20-35, 35-50, over 50, and unknown. For each age group and gender, speakers were allocated to train, dev, and test in a 3:2:5 ratio by utterance count. Consequently, the number of speakers designated for the train, dev, and test sets are 73, 43, and 95, respectively. Additionally, audio from singing and the other scenarios are segregated into an unseen test (**unseen**) set. Therefore, we can not only detect fake utterances on the test set, but also to evaluate the generalization of fake audio detection models on unseen scenarios.

## 4.2 Dataset Description

There are four sets in our impersonation dataset IPAD: train, dev, test and unseen test (**unseen**). Key statistics for different subsets of the impersonation dataset, categorized into "Real" and "Fake" with a further cumulative "Total", are summarized in Table 3.

As our IPAD dataset is partitioned into train, dev, and test subsets based on the dubbing and conversational speech scenarios, with singing and other scenarios being treated as unseen set. In Tables 4, we have meticulously compiled the number of utterances, the count of speakers, and the distribution of speaker profiles — ages, jobs, and hometowns for male and female within each specific scenario for the various subsets.

## 5 EXPERIMENTS

In this section, we first introduce our evaluation metric in Sec. 5.1. In the remaining subsections, we primarily address the following three questions:

- Can models trained on existing fake audio detection dataset ASVspoof2019 LA reliably detect impersonation audio in Sec. 5.2 ?
- How do existing models perform on the impersonation audio dataset IPAD in Sec. 5.3 ?
- Does integrating speaker profiles improve performance on the IPAD dataset in Sec. 5.4?

## 5.1 Evaluation Metric

Equal error rate (EER) is used as the evaluation metric for the detection tasks. Previously, EER is used as the evaluaion metrics for fake audio detection tasks in the ASVspoof [31] and ADD challenges [40, 41]. Let $P_{fa}(\theta)$ and $P_{miss}(\theta)$ denote the false alarm and miss rates at threshold $\theta$ respectively.

$$P_{fa}(\theta) = \frac{\#\{fake\ trials\ with\ score > \theta\}}{\#\{total\ fake\ trials\}} \quad (9)$$

$$P_{miss}(\theta) = \frac{\#\{genuine\ trials\ with\ score < \theta\}}{\#\{total\ genuine\ trials\}} \quad (10)$$

The functions $P_{fa}(\theta)$ and $P_{miss}(\theta)$ monotonically decrease and increase, respectively, as a function of $\theta$. The EER corresponds to the threshold $\theta_{EER}$ at which the two detection error rates are equal,

i.e., EER $= P_{fa}(\theta_{EER}) = P_{miss}(\theta_{EER})$. A lower EER value indicates a model with better performance.

## 5.2 Performance of models trained on ASVspoof2019 LA dataset

*5.2.1 Experimental Setup.* We evaluate the discriminative performance of different combination of frond-end features and back-end classifiers, trained with ASVspoof2019 LA [21] on our IPAD dataset. We choose the ASVspoof2019 LA dataset [21] because it is the most commonly used dataset in fake audio detection research. Our objective is to evaluate whether models trained on this dataset can effectively handle impersonation-type spoofing attacks.

The handcrafted features analyzed include linear frequency cepstral coefficients (LFCC), mel-frequency cepstral coefficients (MFCC), inverted MFCC (IMFCC) and constant-Q cepstral coefficients (CQCC). LFCC is obtained using linear triangular filters. while MFCC originates from mel-scale triangular filters, designed with a denser distribution in lower frequencies to mimic the human ear's perception. IMFCC employs triangular filters arranged linearly across an inverted-mel scale, thereby giving higher emphasis to the high-frequency areas. CQCC is obtained from the discrete cosine transform of the log power magnitude spectrum derived by constant-Q transform. For all these features, we apply a 50ms window size with a 20ms shift and extract features with 60 dimensions. The self-supervised feature includes Wav2Vec 2.0 [3] which combines contrastive learning with masking and HuBERT that uses quantized MFCC features as targets learned with classic k-mean. We leverage the "wav2vec2-base[11]" and "hubert-base[12]" checkpoint from Huggingface's Transformer library [32].

We choose Light CNN (LCNN) [33], Squeeze-and-Excitation network (SENet) [12], Xception [24] and ResNet [10] as our back-end classifiers due to their popularity and effectiveness. Short introductions of these classifiers are provided below.

- LCNN [33] consisting of convolutional and max-pooling layers with Max-FeatureMap (MFM) activation is extensively used as the baseline model of the ASVspoof [31] and ADD [40, 41] competitions.
- SENet [12] dynamically adjusts channel-wise features by explicitly modeling the interdependencies between channels.
- Xception [24], which is employed as a baseline model in [16], utilizes depth-wise separable convolutions to effectively capture both cross-channel and spatial correlations.
- ResNet [10] is introduces as a classifier for fake audio detection in [2], employing a residual mapping.

We also evaluate the performances of four widely used end-to-end competitive models: RawNet2 [14], Raw PC-DARTS [8], Rawformer [37] and AASIST [13] on our proposed dataset. The four end-to-end models are trained on ASVspoof. Brief descriptions are provided below.

- RawNet2 [14] operates directly on raw audio via time-domain convolution. Tak et al. [26] applied it for anti-spoofing, securing second against A17 attacks in ASVspoof 2019.

---

[11]https://huggingface.co/facebook/wav2vec2-base
[12]https://huggingface.co/facebook/hubert-base-ls960

| Features | LA 2019 Test | | | | IPAD Test | | | | IPAD Unseen | | | |
|---|---|---|---|---|---|---|---|---|---|---|---|---|
| | LCNN | SeNet | Xception | ResNet | LCNN | SeNet | Xception | ResNet | LCNN | SeNet | Xception | ResNet |
| LFCC | 3.97 | 3.38 | 2.83 | 4.62 | 48.97 | 57.04 | 58.25 | 59.06 | 63.19 | 46.77 | 52.06 | 63.85 |
| MFCC | 8.23 | 8.06 | 9.02 | 8.83 | 51.57 | 44.03 | 56.75 | 53.82 | 42.88 | 48.62 | 45.24 | 47.38 |
| IMFCC | 21.74 | 24.75 | 16.64 | 12.86 | 48.78 | 57.42 | 56.75 | 57.19 | 50.58 | 46.05 | 43.59 | 40.71 |
| CQCC | 12.39 | 16.84 | 17.45 | 17.64 | 55.61 | 54.48 | 55.26 | 58.87 | 52.51 | 48.62 | 48.47 | 55.98 |
| wav2vec2-base | 1.49 | 1.61 | 1.13 | 1.26 | 56.99 | 45.27 | 57.10 | 62.28 | 63.58 | 45.29 | 43.99 | 58.84 |
| hubert-base | 7.59 | 6.89 | 5.77 | 7.41 | 61.43 | 60.67 | 60.39 | 65.29 | 55.31 | 62.90 | 44.99 | 57.17 |

Table 5: The performances of representative combination of front-end features and back-end classifiers are evaluated on the ASVspoof2019 LA test set, test and unseen set of the IPAD dataset in terms of the EER(%) ↓. The back-end classifiers are trained with ASVspoof2019 LA[21]. LA 2019 Test represents the ASVspoof2019 LA test set.

| Features | IPAD Test | | | | $Avg_{test}$ | IPAD Unseen | | | | $Avg_{unseen}$ |
|---|---|---|---|---|---|---|---|---|---|---|
| | LCNN | SeNet | Xception | ResNet | | LCNN | SeNet | Xception | ResNet | |
| LFCC | 25.37 | 26.48 | 25.03 | 24.99 | 25.46 | 29.89 | 28.18 | 28.90 | 31.15 | 29.53 |
| MFCC | 25.03 | 27.18 | 26.06 | 25.77 | 26.01 | 30.88 | 29.42 | 29.17 | 30.25 | 29.93 |
| IMFCC | 32.74 | 30.05 | 31.36 | 30.12 | 31.06 | 36.62 | 34.12 | 34.38 | 32.00 | 34.28 |
| CQCC | 26.98 | 27.08 | 26.97 | 26.72 | 26.93 | 30.12 | 29.53 | 31.06 | 32.32 | 30.76 |
| wav2vec-base | **23.43** | 23.67 | **23.12** | 23.83 | 23.51 | **27.38** | 28.38 | 30.34 | **28.27** | 28.59 |
| hubert-base | 24.01 | **23.57** | 24.28 | **23.68** | 23.88 | 29.89 | **27.74** | **28.30** | 28.98 | 28.72 |

Table 6: The EER (%) ↓ for different combinations of front-end features and back-end classifiers, assessed on test and unseen subsets of IPAD dataset. The back-end classifiers are trained using our IPAD dataset. The highest result of each classifier is bolded. $Avg_{test}$ and $Avg_{unseen}$ represents the EER (%) ↓ averaged across all back-ends for the test and unseen set, respectively.

| End-to-end Models | LA 2019 Test | IPAD Test | IPAD Unseen |
|---|---|---|---|
| AASIST | 0.83 | 47.03 | 47.26 |
| RawNet2 | 4.59 | 42.01 | 68.40 |
| Raw PC-DARTS | 2.49 | 49.86 | 52.01 |
| Rawformer | 1.15 | 43.27 | 60.77 |

Table 7: The EER (%) ↓ for several classic end-to-end models on the ASVspoof2019 LA test set, test and unseen sets of IPAD dataset. These end-to-end models are trained on ASVspoof2019 LA. LA 2019 Test represents the ASVspoof2019 LA test set.

- Raw PC-DARTS [8] utilizes an automatic approach, which not only operates directly upon the raw speech signal but also jointly optimizes of both the network architecture and network parameters.
- Rawformer [37] integrates convolution layer and transformer to model local and global artefacts and relationship directly on raw audio.
- AASIST [13], which employs a heterogeneous stacking graph attention layer to model artifacts across temporal and spectral segments.

*5.2.2 Experimental Results.* We report the EER (%) for front-end features combined with classifiers and end-to-end models, all trained on ASVspoo2019 LA, across ASVspoof2019 LA test set, IPAD's test set, and unseen set, as detailed in Table 5 and 7.

Results from Table 5 and 7 reveal that models trained on the ASVspoof2019 LA and tested on IPAD dataset exhibit markedly high EER (%), hovering around 50. This is not surprising, as ASVspoof2019 is mainly tailoring for identifying machine-generated audio and real audio. In contrast, our IPAD dataset comprises solely of human-produced audio, resulting in failure detection of impersonation audio when models are trained on ASVspoof2019 LA. This indicate that models trained on the ASVspoof2019 LA struggle to detect impersonated audio, suggesting that impersonation as an attack type can significantly increase the success rate of spoofing attacks.

## 5.3 Performance of models trained on the IPAD dataset

*5.3.1 Experimental Setup.* The front-end features combined with classifiers and end-to-end models are the same as those evaluated in Sec. 5.2, but trained on the IPAD's train set.

*5.3.2 Experimental Results.* For handcrafted features combined with classifiers, from Table 6, we observe that, with the exception of the LFCC+LCNN combination on the test set and the LFCC+ResNet on the unseen set, the LFCC feature generally outperforms other handcrafted features, suggesting its effectiveness in impersonation audio detection.

As presented in Tables 6, our findings reveal that self-supervised features outperform handcrafted features on both the test and unseen sets. Specifically, averaged over four different back-ends, wav2vec-base exhibits an average performance of 23.51 and 28.59 on IPAD's test and unseen sets, respectively. However, the best hand-crafted feature demonstrates performance of 25.46 and 29.53 on IPAD's test and unseen sets. This indicates that pretrained models are more adept at capturing information pertinent to impersonation audio detection compared to handcrafted features.

| End-to-end Models | IPAD Test | IPAD Unseen |
|---|---|---|
| RawNet2 | 27.44 | 31.19 |
| Raw PC-DARTS | 26.66 | 37.16 |
| Rawformer | 28.44 | 35.12 |
| AASIST | **23.73** | **30.25** |

Table 8: The EER (%) ↓ for several representative end-to-end models on both IPAD's test and unseen sets. These end-to-end models are trained on IPAD's train set. The highest result of each subset is bolded.

| | IPAD Test | | IPAD Unseen | |
|---|---|---|---|---|
| | w/o | w/ | w/o | w/ |
| wav2vec2-base | 23.43 | 22.78 | 27.38 | 25.13 |
| wav2vec2-large | 24.49 | 23.62 | 33.98 | 30.97 |
| hubert-base | 24.01 | 22.89 | 29.89 | 25.04 |
| hubert-large | 24.77 | 23.04 | 30.34 | 25.49 |
| wavlm-small | 24.36 | 22.56 | **27.08** | 26.54 |
| wavlm-large | **23.28** | **21.97** | 27.49 | **23.96** |

Table 9: The performance is evaluated on test and unseen set of the IPAD dataset in terms of the EER(%) ↓. w/o and w/ represents whether speaker profiles are integrated.

Moreover, when averaging features, the ResNet backend achieves the lowest EER (%) 25.85 on the test set, showing the best performance, while SeNet exhibits superior generalization with the lowest EER (%) 29.56 on the unseen set.

For end-to-end models, Table 8 reveals that among the evaluated models, AASIST achieves the best performance on both the test and unseen set with the lowest EER (%) at 23.73 and 30.25, respectively, demonstrating superior performance in impersonation audio detection and great generalization capabilities in unseen conditions.

In this subsection, we evaluate the performance of existing models on the IPAD dataset. The results indicate that there is still significant room for improvement.

## 5.4 Performance of our proposed speaker profiles integrated method

*5.4.1 Experimental Setup.* As indicated in Sec 5.3, self-supervised features outperform handcrafted features in the detection of impersonation audio, thus we employ self-supervised features in our speaker profiles integrated method. The self-supervised models considered include Wav2Vec 2.0 [3], HuBERT [11] and WavLM [5]. WavLM [5], largely paralleling HuBERT, introduces advancements in spoken content and speaker identity by integrating a gated relative position bias and enriching training data with an utterance mixing approach.

We utilize the self-supervised models as the frond-end feature extractor and instance the Embedding layer of the speaker profile extractor as the convolutional waveform encoder of the corresponding frond-end feature extractor. The output size of the speaker profile extractor in Sec. 3.2 is 128 in our experiments.

For self-supervised pre-trained models, we leverage the pre-trained checkpoint from Huggingface's Transformer library [32]. Below are the models used in our experiment: "wav2vec2-base[13]", "wav2vec2-large[14]", "hubert-base[15]", "hubert-large[16]","wavlm-base"[17] and "wavlm-large" [18]. For the back-end classifier, we opted for the widely utilized LCNN [33].

*5.4.2 Experimental Results.* We report the model's performance on the IPAD dataset with ("w/") and without ("w/o") speaker profile

---

[13]https://huggingface.co/facebook/wav2vec2-base
[14]https://huggingface.co/facebook/wav2vec2-large
[15]https://huggingface.co/facebook/hubert-base-ls960
[16]https://huggingface.co/facebook/hubert-large-ll60k
[17]https://huggingface.co/microsoft/wavlm-base
[18]https://huggingface.co/microsoft/wavlm-large

information. Results for test and unseen sets of IPAD are detailed in Table 9.

We observe that wavlm-large consistently yields the best results on both the test and unseen sets, when no speaker profiles are integrated, indicating its robust and useful audio feature extraction capabilities. Surprisingly, we find that the performance of wav2vec-large is inferior to that of wav2vec-small, and similarly, hubert-large underperforms hubert-small. We speculate that this may be due to the fact that the models were only pretrained on genuine audio, and simply increasing model size does not enhance the model's ability to detect impersonation audio. Furthermore, when comparing base models, wav2vec-small emerges with the optimal performance.

We find that the incorporation of speaker profiles can significantly enhances the detection of impersonation audio. wavlm-large with speaker profile information integrated achieved the best EER(%) of 21.97 and 23.96 on IPAD's test and unseen sets, respectively. The inclusion of speaker profiles has led to notable improvements for all six pretrained models on both IPAD's test and unseen sets. On average, the EER(%) decreased by 1.26 on the test set and by 3.17 on the unseen set. This suggests that the utilization of speaker profiles enable models to better leverage information such as the speaker's job and age in the detection of imitated audio. Additionally, the more pronounced improvement on the unseen set indicates that the introduction of speaker profiles bolsters the model's generalization capabilities in out-of-domain situations.

## 6 CONCLUSIONS

In this work, we investigate countermeasures against impersonation. Different from spoofing attacks like TTS and VC, which leave physical or digital traces, impersonation involves live human beings producing entirely natural speech. We propose a novel strategy that utilize speaker profiles for impersonation audio detection. Moreover, we propose the ImPersonation Audio Detection (IPAD) dataset to promote the community's research on impersonation audio detection, filling the gap that there is no large-scale impersonated speech corpora available. To provide baselines for future practitioners, we train several existing models on our IPAD dataset. Finally, we demonstrate that incorporating speaker profiles into the process of impersonation audio detection can achieve notable improvements. Future work includes constructing an English-language impersonation dataset and exploring how to better utilize speaker profiles from other modalities for impersonation audio detection.

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
