# OpenReview forum: "Utilizing Speaker Profiles for Impersonation Audio Detection"
_acmmm.org/ACMMM/2024/Conference — MM2024 Poster_

### Official Review · Reviewer_zGae · 2024-05-04

**Rating:** 5
**Confidence:** 3

**Summary:**

Fake audio detection is an emerging active topic. A growing number of literatures have aimed to detect fake utterance,which are mostly generated by Text-to-speech(TTS) or voicecon version(VC).However, countermeasures against impersonation remain an underexplored area.This paper presents the first diverse-scenarios, diverse-speaker impersonation dataset, named ImPersonation Audio Detection (IPAD), to promote the community’s research on impersonation audio detection. Also, propose a novel method that integrates speaker profiles into the detection of impersonation audio. I believe this paper has significant practical implications for the future development of audio detection. Therefore, I am inclined to accept this paper.

**Strengths:**

1. The paper is well-written and clear.
2. It highlights the lack of research on impersonation issues in existing audio spoofing detection and constructs a dataset based on impersonation.
3. It introduces a graph-based method to extract speaker profiles, enhancing the model's performance.

**Limitations:**

1. My first question is any voice that exhibits impersonation considered "fake," or does it specifically need to mimic a target label already in the dataset?
2. I recommand the authors conduct a small experiment to verify whether the impersonation voices in the dataset can be misclassified by neural networks, similar to TTS and VC audio.
3. The paper lacks a demo that could showcase some audio samples from the current database.
4. I would like the authors to present the results of four end-to-end models in Section 5.4.
5. The main results of the experiments focus on the ASVspoof2019 dataset, which needs to be described in Table 1.

**Suitability:**

2

---

### Official Review · Reviewer_rAAZ · 2024-05-14

**Rating:** 2
**Confidence:** 4

**Summary:**

this manuscript proposes a novel method that integrates speaker profiles into the process of impersonation audio detection. Speaker profiles are inherent characteristics that are challenging for impersonators to mimic accurately, such as the speaker's age, job. the authors aim to leverage these features to extract discriminative information for detecting impersonation audio. Moreover, there is no large impersonated speech corpora available for quantitative study of impersonation impacts. To address this gap, the authors further design the first large-scale, diverse-speaker Chinese impersonation dataset, named ImPersonation Audio Detection (IPAD), to advance the community's research on impersonation audio detection. the authors evaluate several existing fake audio detection methods on proposed dataset IPAD, demonstrating its necessity and the challenges. Additionally, the findings reveal that incorporating speaker profiles can significantly enhance the model's performance in detecting impersonation audio.

**Strengths:**

1：authors propose a novel method that integrates speaker profiles into the detection of impersonation audio. To this end, they utilize a graph-based approach to extract speaker profile information. Additionally, the proposed method does not require labeled speaker profiles during the test period.


2：authors present the first diverse-scenarios, diverse-speaker impersonation dataset, named ImPersonation Audio Detection (IPAD), to promote the community's research on impersonation audio detection.

**Limitations:**

1:I am concerned about the fairness of this experiment. Taking Table 5 as an example, the compared dataset contains real samples and deepfake samples, while the proposed dataset is filled with real samples, which inevitably leads to the failure of existing detection methods.

2. This manuscript lacks relevant work on non-personalized audio detection, and we hope to supplement the relevant knowledge background.

3:I am concerned about the practicality of this scenario. In most cases, the detector only has one voice clip and no prior knowledge of the speaker. In addition, the speaker's healthy condition, age, and recording time can all have impact on the speaker's timbre, and these data are difficult to track.

**Suitability:**

2

---

### Official Review · Reviewer_hDX5 · 2024-05-18

**Rating:** 4
**Confidence:** 3

**Summary:**

In this paper, the authors proposed a new speech signal classification problem known as impersonation detection, where a classifier is asked to determine whether a speech signal is recorded from a human subject or a human object that is impersonating someone else.

Due to the lack of existing datasets, the authors proposed a dataset for this purpose known as IPAD.

The authors proposed a method for impersonation detection which is trained on IPAD. The proposed method is not only based on low-level analysis of the speech signal but also high level analysis such as the deducing the speaker's gender, age, origin, etc.

The proposed method achieved higher performance than other methods that are trained for synthetic speech detection tasks.

**Strengths:**

- The authors proposed the new IPAD dataset
- The authors proposed a new method for the IPAD dataset, where the speaker attribute is fused with audio features extract from waveform for analysis
- The proposed method achieved better performance on the IPAD dataset compared to existing methods designed for synthetic speech detection

**Limitations:**

- The performance gap of the proposed method and methods trained on ASVspoof2019 is not very big. Why do methods designed for ASVspoof2019 work on impersonation detection problem?  The methods designed for ASVspoof2019 tend to operate on very short speech signals, and it is believed that they rely on low-level features to make decisions. If they also work for IPAD dataset, I think there can be two possibilities:
  1. Low-level features alone can be used for impersonation detection. However, since all signals in IPAD are human speech signals, it may be difficult to classify them based on low-level features only.
  2. The diversity of IPAD dataset is limited. That is, all methods trained on IPAD tend to overfit the dataset. The performance evaluation results are not reliable.

- In IPAD dataset, the authors only include signals from experienced impersonators. It would be interesting for authors to report the performance of the the proposed method and baseline methods on:
  - Synthesized impersonation speech
  - Impersonation speech from less skilled impersonators

**Suitability:**

3

---

### Official Review · Reviewer_4xtp · 2024-05-23

**Rating:** 5
**Confidence:** 3

**Summary:**

Thank you to all the authors for their efforts and for the opportunity to review the manuscript. The paper presents an approach for impersonation audio detection that uses integrated speaker profiles such as their job, location, gender, etc. The authors also develop and plan to release the ImPersonation Audio Detection (IPAD) dataset for fostering further research in the area. Comprehensive experiments demonstrate the validity of the hypothesis.

**Strengths:**

1. The paper proposes a novel approach to incorporate speaker profile information to enhance detection of impersonated audio.
2. The introduction, besides minor typos is well-written. The authors carefully distinguish generated audio from natural impersonated speech describing the motivation and their contributions clearly.
3. Figure 1 is very illustrative and useful for demonstrating the training process. However, the font size is too small. If font size can be increased, it would be more presentable.
4. The experiments and results are significant enough to prove the correctness of authors’ hypothesis and to demonstrate the usefulness of IPAD Dataset. However, improvements can be made by experimenting with performance on cross datasets. For example, how do all the baselines trained on IPAD Dataset perform when evaluated on old impersonation datasets such as Hautamäki et al. [9] and Lau et al. [18]. This could be provided as an additional experiment in the supplementary material.

**Limitations:**

1. Section 2.1 is well-written but includes too many details about fake audio detection (synthetic) but not much detail about impersonated or mimicked audio (real) which is the main focus of the paper. Some efforts have explored methods in this direction and should be referenced to provide readers with comprehensive information. Some examples that could be included in references (not comprehensive or exhaustive list):
    1. https://ieeexplore.ieee.org/document/9215407
    2. https://www.cs.joensuu.fi/odyssey2014/program/pdfs/54.pdf
    3. https://link.springer.com/article/10.1007/s10772-012-9163-3
2. “The corruption function used here is designed to encourage the representations to properly encode structural similarities of different nodes.” Could you explain in slightly greater detail how is the corruption function helping the proposed method? In text, it is clear how it is implemented but not clear how it helps enhance performance. Perhaps the explanation could include some experimental validation.
3. What is the value of s used in experiments? Is it same as the number of speaker attributes, i.e. name, hometown, age, job, gender, and the scenario?

Besides, there are some minor grammatical errors and typos which could be removed by proof-reading:

1. Abbreviations should preferably be used the first time the phrase is mentioned. For example, TTS should be defined in line 65 instead of line 66. Similarly for VC. This is very minor but will improve the writing of the manuscript.
2. “Partially fake focuses on only changing several words in an utterance. where fake segment is generated by manipulating the original utterances with genuine or synthesized audio clips.” If this is a full sentence, there should be a ‘,’ instead of a ‘.’
3. In line 83, “dialect,lexical”, there is a missing space.
4. Line 73 - full form should also be mentioned for ADD.
5. Line 87, “Wu et al.” should be treated as a plural subject instead of singular.
6. Line 137 - “Hautamäki et al. [9] constructs a small Finnish impersonation dataset in 2013.” “constructs” should be “constructed”.
7. Line 145 - “pratical” should be spelt correctly as “practical”.
8. Line 173 - “previous can be roughly divided into two categories”, there is a “work” missing after “previous”.
9. Consistency in full-form of abbreviations, for example, in line 187, Linear frequency cepstral coefficients, either all first letters should be uppercase throughout the manuscript or this should be made consistent with other full-forms.
10. “In 2004, an impersonation database is” should be past tense with the use of “was”.
11. Line 330 - “We first give a introduce to the problem statement.” “a introduce” should be “an introduction”

There are several more which I could not list here. A proof-read and grammar check is strongly recommended.

Overall, the proposed method is novel and significant analysis is present in the paper. However, major improvements need to be made in writing, such as correct grammar and use of correct scientific terminology as described above.

**Suitability:**

3

---

### Meta-Review · Area_Chair_6taB · 2024-07-04

**Recommendation:** Accept (Poster)
**Confidence:** 5

**Metareview:**

The paper addresses the problem of fake audio detection and proposes a method that integrates speaker profiles into the process of impersonation audio detection. It also introduces a large-scale, diverse-speaker Chinese impersonation dataset, named ImPersonation Audio Detection (IPAD). Presents the performance of selected existing fake audio detection methods on the proposed dataset IPAD.

The proposed method employs speaker profile (such as speaker's age and job) in audio detection and utilizes a graph-based approach to extract it; which is a limited novelty as textual (semantic) information is used with similar approaches in audio-related tasks in recent work. On the other hand, overall, constructing a new large-scale dataset and evaluating it with selected methods put a value on it. After considering the paper, the reviewer's comments, and the rebuttal I recommend 'accept (poster)' for the paper.

The reviewers highlight the following strengths and limitations:

Strengths:
1. The paper proposes a novel approach to incorporate speaker profile information to enhance the detection of impersonated audio.
2. The experiments and results are significant enough to prove the correctness of the authors’ hypothesis and to demonstrate the usefulness of IPAD Dataset.
3. The paper is well-written and clear.

Limitations:
1. A description of the ASVspoof2019 dataset is missing.
2. The paper lacks the results of 4 end-to-end system performances when speaker profiles are integrated.